# Novel Nematocidal Compounds from Shrimp Shell Wastes Valorized by *Bacillus velezensis* RB.EK7 against Black Pepper Nematodes

Thi Huyen Trang Trinh [1], San-Lang Wang [2,3,*], Van Bon Nguyen [4,*], Tu Quy Phan [1], Manh Dung Doan [4], Thi Phuong Hanh Tran [1], Thi Huyen Nguyen [4], Thi Anh Hong Le [5], That Quang Ton [6] and Anh Dzung Nguyen [4,*]

1   Department of Science and Technology, Tay Nguyen University, Buon Ma Thuot 630000, Vietnam
2   Department of Chemistry, Tamkang University, New Taipei City 25137, Taiwan
3   Life Science Development Center, Tamkang University, New Taipei City 25137, Taiwan
4   Institute of Biotechnology and Environment, Tay Nguyen University, Buon Ma Thuot 630000, Vietnam
5   Institute of Tropical Biology, Vietnam Academy of Science and Technology, Ho Chi Minh City 700000, Vietnam
6   Faculty of Chemistry, University of Science, Ho Chi Minh City 700000, Vietnam
*   Correspondence: sabulo@mail.tku.edu.tw (S.-L.W.); nvbon@ttn.edu.vn (V.B.N.); nadzung@ttn.edu.vn (A.D.N.)

**Abstract:** Among various organic wastes, shrimp shell powder (SSP) was the most suitable carbon/nitrogen source for producing antinematode compounds (ANCs) via *Bacillus veleznesis* RB.EK7 fermentation. The fermentation process for the enhancement of antinematode activity was investigated. *B. veleznesis* RB.EK7 produced the highest antinematode activity in the medium containing 0.8% SSP with an initial pH of 6.5–7.0, and fermentation was performed at 35–37 °C with a saking speed of 150 rpm for 72 h. Targeting ANCs were purified from the fermented culture broth and identified as thymine (**1**) and hexahydropyrrolo [1,2-a]pyrazine-1,4-dione (**2**) based on nuclear magnetic resonance (NMR) and mass spectra analysis and were compared to those of the reported compounds. Notably, for the first time, these compounds were found as novel ANCs. Thymine (**1**) demonstrated a potential nematicidal effect with near 100% mortality of second-stage juvenile (*J2*) nematodes and anti-egg hatching effects of 70.1%, while hexahydropyrrolo [1,2-a]pyrazine-1,4-dione showed moderate antinematode activities with 64.2% mortality of *J2* nematodes and anti-egg hatching effects of 57.9%. The docking study coupled with experimental enzyme inhibition results indicated that the potent nematicidal effect of these compounds may be possibly due to the inhibition of the targeting enzyme acetylcholinesterase. The data of this study suggest that SSP can be potentially reused for the eco-friendly production of ANCs for the management of black pepper nematodes.

**Keywords:** black pepper; *Bacillus veleznesis*; nematodes; antinematode compounds; organic wastes; microbial fermentation; thymine

## 1. Introduction

Black pepper is one of the important industrial crops with high economic value for export. Its product, peppercorn, has been considered a common daily spice and is the most widely traded spices reaching around 20% of all spices traded commercially [1,2]. This crop is widely planted in Vietnam, Indonesia, India, and Brazil. Vietnam is the largest producer and exporter of peppercorns, with approximately 40% of 546,000 tons worldwide produced [3]. However, the cultivation of this spicy plant faces various pathogen diseases, including the root–knot nematode [4], and one of the major nematodes that seriously damages black pepper is the *Meloidogyne incognita* species [5].

Up until now, many methods have been investigated for the management of root–knot nematode, including using chemical nematicides, biological control, chemicals, cultivars in cotton in a semi-arid environment, co-cultivation with various plants, destroying infected

plants, resistant planting materials, and nanotechnology [6–9]. In current years, for eco-friendly controlling nematodes and the benefit of agricultural production, various beneficial active microbes have been extensively studied [10,11]. Chemical nematicides are still the most effective means to manage nematodes. However, the long-term use of nematicide compounds such as carbamate and organophosphorus has led to increased nematode resistance, including a lack of effective field control and environmental pollution. Thus, there is a constant search for a new source of nematicidal compounds [6].

Nematicides may be obtained from chemical synthesis or natural sources such as plants and microbial fermentation [5,6,12–14]. Recent studies indicated that various antinematode compounds of bacterial and fungal origins such as prodigiosin [5], hemi-pyocianin [14], and 2-furoic acid [13] could serve as a preferable lead structure for novel nematicide research and development. The use of nematicides from chemical synthesis may cause environmental issues, reducing the quality of agro-products and the soil microbial diversity, as well as increasing the resistance of nematodes [12,15,16]. Thus, to reduce the toxicity and other problems, various natural compounds have been investigated for the potential management of nematodes [15]. Among the natural resources, the identification and production of antinematode compounds (ANCs) from microbes has received much attention since they may be produced on an industrial scale for available demand [5]. In current microbial fermentation science, the aspect of reusing organic wastes for the cost-effective production of valuable compounds is an emerging research topic [17–20]. However, the eco-friendly methods for the production of ANCs from organic wastes have been reported in very few literature studies [5,14].

Considering the green and effective treatment of root–knot nematodes, in our earlier study, we isolated and screened the beneficial microbes with an antinematode effect in the Central Highlands of Vietnam [21], and *B. veleznesis* RB.EK7 was found as a novel potent anti-nematode bacterial strain. In this study, the antinematode activity of *B. veleznesis* RB.EK7 was significantly enhanced via the optimization of culture conditions using organic wastes as the C/N source for fermentation. The major ANCs were extracted and purified, and their chemical structures were elucidated. The docking study and acetylcholinesterase inhibitory activity were also performed to investigate the molecular mechanism of the anti-nematode activity of the ANCs.

## 2. Materials and Methods

### 2.1. Materials

Rhizobacterial strain *B. velezensis* RB.EK7 was obtained from our previous work [21]. Eggs of *Meloidogyne* sp. were isolated from their galls of the nematode on sick black pepper plants cultivated in Buon Ma Thuot city, which were then used for the preparation of *J2* nematodes. Shrimp shell powder (SSP) was obtained from Shin-Ma Frozen Food Co. (I-Lan, Taiwan), and some other organic wastes, including cassava residue waste (CRW), peanut oil processing by-product (groundnut cake, GC), and soybean residue waste (SRW), were obtained in Buon Ma Thuot city, Dak Lak province, Vietnam. Acetylcholinesterase, yeast extract, peptone, and silica gel (Geduran® Si 60, size: 0.040–0.063 mm) were purchased from Sigma Chemical Co. (St. Louis City, MO, USA), and some solvents used in this study were from Sigma Aldrich.

### 2.2. Methods

#### 2.2.1. Production of Antinematode Compounds by *B. velezensis* RB.EK7 Fermentation

Screening Organic Wastes as C/N sources for fermentation:

Various organic wastes, including soybean residue, shrimp shell powder (SSP), cassava residue (CR), groundnut oil processing (GOP), and a commercial medium (yeast extract/peptone = 5/3) were used as C/N (0.8%) sources for fermentation. The salt composition of the medium included 0.6% $KH_2PO_4$ and 0.4% $(NH_4)_2SO_4$. The culture medium was sterilized by autoclaving at 121 °C for 15 min. Fifty milliliters of the medium in a 250 mL Erlenmeyer flask was inoculated with 1 mL bacterial seed and fermented at 37 °C

in a rotary shaker operating at 150 rpm for 72 h. The supernatant was harvested by centrifugation at 13,000 rpm for 15 min and then used for estimation of antinematode activity. The shrimp shell powder was found to be the most suitable substrate to produce ANCs and was, thus, further used for further experimentation.

The effect of some culture conditions on antinematode compounds production:

To enhance antinematode activity, some factors, including initial pH (5.0, 5.5, 6.0, 6.5, 7.0, 7.5, 8.0, 8.5, and 9.0), cultivation temperature (29, 31, 33, 35, 37 and 39 °C), shaking speed (0, 100, 150, 200, and 250 rpm), and cultivation time (24, 48, 72, 96, and 120 h), were examined for their effects. The supernatant was harvested by centrifugation at 13000 rpm for 15 min and then used in the estimation of antinematode activity. The following experiments were designed based on the optimal conditions achieved from previous experiments.

### 2.2.2. Purification and Identification of Antinematode Compounds and Their Chemical Structures

The culture supernatant (CS) of strain RB.EK7 was harvested by centrifugation at 13,000 rpm for 15 min. CS evaporated at 50 °C to dried powder (199 g). Then, it was extracted by solid–liquid with solvent systems, including n-hexane, chloroform, ethyl acetate, ethyl acetate: methanol (90:10), ethyl acetate: methanol (60:40), and methanol. The extracts were obtained and evaporated under low pressure to recover the solvent, obtaining high grades. The crude extract was used to evaluate antinematode activity. The extracted phases of ethyl acetate: methanol (90:10) showed high nematicidal activity and mass. Thus, this extract was further used for the isolation of the ANCs.

Extracts with chloroform and ethyl acetate were mixed and further separated via a silica gel opened column and eluted with $CH_2CL_2$-MeOH (95:5, *v/v*) to obtain compound **1** (60.2 mg). The extract of ethyl acetate: methanol (90:10) was loaded onto a silica gel opened column and eluted with $CH_2CL_2$-MeOH (9:1, *v/v*), $CH_2CL_2$-MeOH (5:5, *v/v*), and MeOH to obtain five subfractions. From subfraction 5.4, compound **2** (28.1 mg) was isolated. The chemical structures of these active compounds were identified based on nuclear magnetic resonance (NMR) and mass spectra analysis and compared to those of the reported compounds. The purity of these purified compounds was confirmed by ultra-high-performance liquid chromatography (UHPLC). The extraction and purification process are summarized in Scheme 1.

### 2.2.3. Nematocidal Activity Assays

Nematocidal activity assays were performed following the methods presented in our earlier publications [5,14]. The sick black pepper roots were collected from Buon Ma Thuot, Dak Lak province, Vietnam, and eggs and *J2* nematodes were prepared according to the method of Khan et al. [22]. The process for the preparation of nematode eggs and *J2* root-knot nematodes from the sick black pepper roots is illustrated in Scheme 2. The samples, including culture supernatants and compounds dissolved in DMSO at various concentrations, were tested for their nematicidal activity via the effect on *J2* nematodes and egg-hatching; then, their activities were expressed as the anti-*J2* inhibition efficiency (%) and egg-hatching rate (%).

The anti-*J2* nematode effect:

Two hundred microliters of sample solution (200 μL) was mixed with 100 μL sterile distilled water containing 30 individuals of *J2* nematodes in a 1 mL Eppendorf tube and kept at 20 °C for 24 h. In the control group, 200 μL DMSO without antinematode compounds was also mixed with 100 μL sterile distilled water containing 30 individuals of *J2* nematodes in a 1 mL Eppendorf tube and kept at the same condition. The number of immobilized nematodes was counted under a stereoscopic microscope Olympus SZ51 (Shinjuku Monolith, 3-1, Nishi Shinjuku 2-chome, Shinjuku-ku, Tokyo, Japan) and used to estimate the activity. All tests were carried out in triplicates.

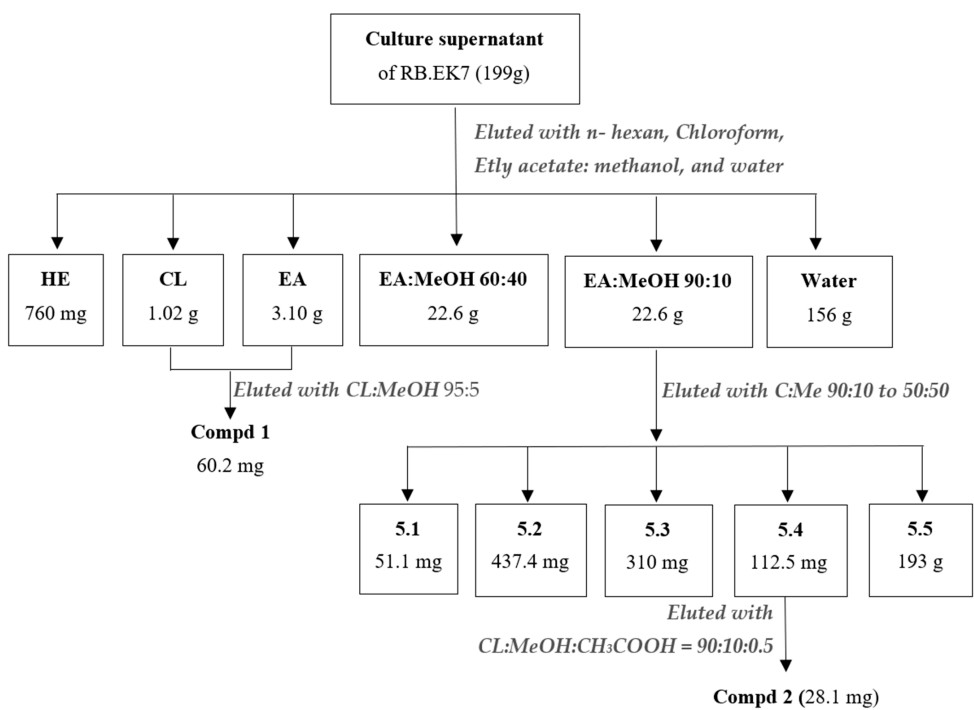

**Scheme 1.** The scheme of extraction and purification of antinematode compounds. HE: hexane; CL: chloroform; EA: ethyl acetate; MeOH: methanol.

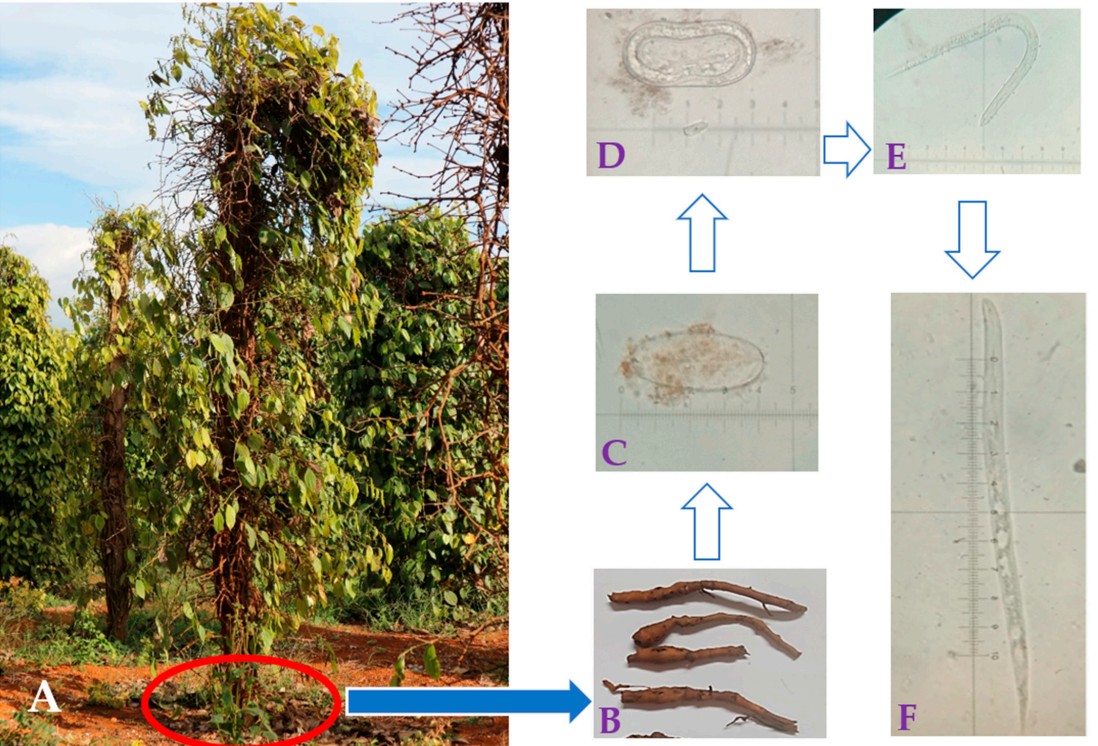

**Scheme 2.** The process for the preparation of nematode eggs and *J2* root–knot nematodes from sick black pepper roots. The sick black pepper with yellow leaves (**A**) was chosen for collecting its root–knots (**B**), which was used for isolation of nematodes eggs (**C**). Nematode eggs were further incubated for 3–5 days for the eggs to hatch (**D**), and *J2* root–knot nematodes were obtained (**E**). *J2* nematodes were immobilized (considered dead nematodes) after treatments with antinematode compounds (**F**).

The egg-hatching inhibitory effect:

Two hundred microliters of the sample solution was mixed with 100 μL sterile distilled water containing 200 nematode eggs in a 1 mL Eppendorf tube, and this mixture was kept at 20 °C for 3 days. In the control group, 200 μL DMSO without antinematode compounds was also mixed with 100 μL sterile distilled water containing 200 nematode eggs in a 1 mL Eppendorf tube and kept at the same condition. The number of hatched eggs was counted based on *J2* nematodes using a stereoscopic microscope Olympus SZ51 (Shinjuku Monolith, 3-1, Nishi Shinjuku 2-chome, Shinjuku-ku, Tokyo, Japan). All tests were carried out in triplicates.

### 2.2.4. High-Performance Liquid Chromatography Analysis of Purified Compounds

The compounds were dissolved in methanol, filtered through a 0.22 μm membrane, and 5 μL of each was injected into the HPLC systems (Thermo-Ultimate 3000 UPLC system—ThermoScientific, Dornierstr. 4 D-82110 Germering Germany) using a column (Hypersil GOLD aQ C18 column, 150 mm × 2.1 mm, particle size 3 μm, at a temperature of 30 °C). The compound thymine was eluted by solvent systems of MeOH (100%)/ammonium acetate (10 mM in water) with a flow rate of 0.8 mL/min, and the UV detection wavelength was 254 nm. The compound hexahydropyrrolo [1,2-a] pyrazine-1,4-dione was eluted by solvent systems of 20% MeOH in water with a flow rate of 0.7 mL/min, and the UV detection wavelength was 210 nm.

### 2.2.5. Docking Study Protocol

The virtual study protocol was performed according to the method presented in our earlier works [17,23,24] with three typical steps as stated below.

Preparation of acetylcholinesterase (AChE) structure and the active sites on AChE:

The protein structure data of AChE were obtained from the Worldwide Protein Data Bank; then, the 3-D form was produced by using MOE-2015.10 software. A virtual pH 8 was set to prepare the structures of protein structure. The active sites on AChE were selected using the site finder function in MOE software after removing all water molecules.

Preparation of ligands (inhibitor compounds):

Ligands structures were prepared and further optimized using the MOE system with parameters of Force field MMFF94x; R-Field 1: 80; cutoff, Rigid water molecules, space group p1, cell size 10, 10, 10; cell shape 90, 90, 90; and gradient 0.01 RMS kcal.mol$^{-1}$A$^{-2}$. A virtual pH 8 was also set for the preparation of the structures of the ligands.

Docking performance of ligands into the active sites on AChE:

The virtual study was performed on ligands with AChE using MOE-2015.10 software, and the output data, including Root Mean Square Deviation (RMSD), docking score (DS), interaction types (linkages), amino acid composition, interaction between amino acids with the binding site in AChE, and the distances of the linkages were obtained for analysis.

### 2.2.6. Achetylcholinesterase Inhibition Assay

Achetylcholinesterase inhibitory activity was tested using Ellman's assay [25] with modifications. Forty microliters of sample solution (compounds at different concentrations) was mixed with 40 μL of AChE solution (0.5 mM) and 80 μL of 0.05 M phosphate buffer, pH 8.0, and kept at at 25 °C for 15 min in a flat-bottom 96-well plate. The reaction started when 20 μL of agent 5, 5′-dithiobis-2-nitrobenzoic acid (DTNB, 0.003 M) and 40 μL of substrate achetylthiocholine iodide (ATCI, 0.002 M) were added. The reaction was kept 25 °C for 10 min; then, absorbance was measured at 415nm (SpectraMax M2, 96-well plate reader). In the control group, 40 μL of 0.05 M phosphate buffer, pH 8.0, was used instead of 40 μL of the sample solution. The enzymatic inhibitory activity was estimated using the below-mentioned equation:

$$\text{AChE inhibitory activity (\%)} = (C - E)/C \times 100,$$

where E is the optical density of the reaction containing sample (inhibitor) and enzyme, while C is the optical density of the reaction containing enzyme and the same volume of 0.05 M phosphate buffer instead of the sample. In this assay, 0.05 M phosphate buffer, pH 8.0, was used to dissolve enzyme AChE, substrate achetylthiocholine iodide (ATCI), and agent 5, 5′-dithiobis-2-nitrobenzoic acid (DTNB). The compounds (thymine and hexahydropyrrolo [1,2-a] pyrazine-1,4-dione) were dissolved in DMSO and eluted at various concentrations using a 0.05 M phosphate buffer.

## 3. Results and Discussion

### 3.1. Production of Antinematode Compounds by Fermentation

C/N sources play an important role in enhancing the yield of secondary metabolites produced by microbes during fermentation [14]. To obtain highly productive ANCs, various organic wastes, including SR, SSP, CR, and GOP, and a commercial medium (CM, yeast extract/peptone = 5/3) were used as C/N (0.8%) sources for fermentation. The results in Figure 1a show that all supernatants in the media containing organic wastes and commercial medium fermented by *B. veleznesis* RB.EK7 displayed antinematode activity in the range of 50.67–98%. Among these organic wastes, SSP was found as the most suitable C/N source for producing ANCs. The supernatant produced by RB.EK7 using SSP as the sole C/N source for fermentation demonstrated the highest nematicidal effect (98%), while the commercial medium produced an activity of 89.67%, and other organic wastes produced lower activities in the range of 50.67–73.67%. Thus, the SSP was chosen for further investigation.

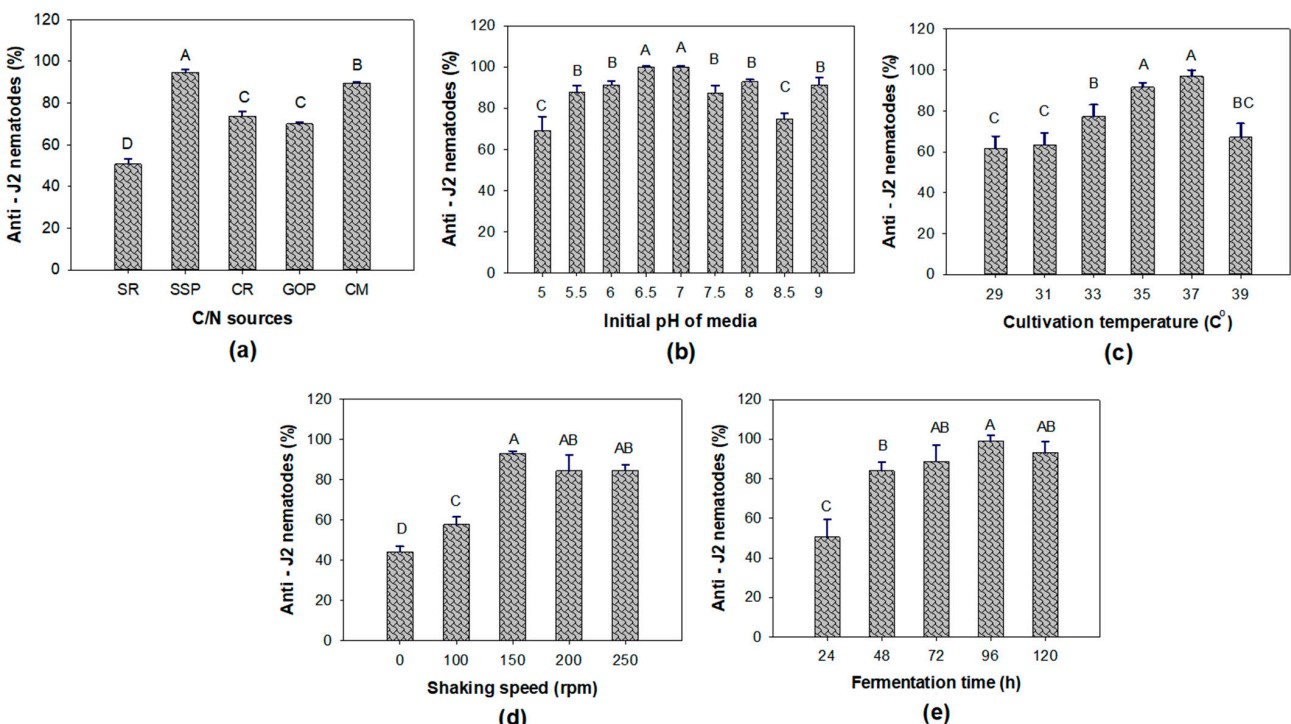

**Figure 1.** The production of antinematode compounds. The effect of C/N sources (**a**), the initial pH (**b**), cultivation temperature (**c**), shaking speed (**d**), and cultivation time (**e**) on the antinematode activity of supernatants produced by *B. veleznesis* RB.EK7 were examined. Values in the same figure with the different letters (A–D) are significantly different.

With respect to the effective use of the organic waste SSP for a high yield of ANCs, some factors, including the initial pH (5.0–9.0), cultivation temperature (29–39 °C), shaking speed (0–250 rpm), and cultivation time (24–120 h), were examined for their effect. Taken together, *B. veleznesis* RB.EK7 produced the highest nematicidal activity when it was

cultivated in the conditions of pH 6.5–7.0 (Figure 1b), cultivation temperature of 35–37 °C (Figure 1c), shaking speed of 150 rpm (Figure 1d), and cultivation time of 72 h (Figure 1e). The bacterial density was also examined during the fermentation process; however, there seemed to be no correlations between culture density and antinematode activity (the data are not shown).

Shrimp shell is one of the most abundant chitinous wastes obtained mainly from by-products of the fishery-processing industry [26]. Through microbial conversion, various other bioactive materials such as proteases, chitinase, chitosanases, and oligomers of chitin and chitosan, as well as antioxidants, anticancer, and antidiabetic agents, have been produced from shrimp shells [20,27–30]. However, this is the first report on the utilization of SSP for the production of ANCs via microbial fermentation.

### 3.2. Purification and Identification of the Chemical Structures of Antinematode Compounds Produced by B. velezensis RB.EK7

Antinematidic agents may be enzymes or natural compounds with small molecules [1]. Thus, for the rapid prediction of active antinematidic agents for purification, we tested the activity of some enzymes related to the antinematode effect (chitinase and protease) and nematicidal activity of supernatants produced by *B. velezensis* RB.EK7. As shown in Table 1, the supernatant demonstrated high chitinase activity (4.5 IU/mL), no protease activity, and high antinematode activity with 93.67% mortality with respect to *J2* nematodes. This result indicated that the nematicidal effect of the RB.EK7 supernatant is not due to protease but may be due to the chitinase effect. To elucidate this result, the supernatant was heated to a high temperature (at 100 °C for 60 min); then, the enzyme and ant nematodes effects were tested. After being treated at 100 °C, the supernatant did not exhibit any chitinase activities; however, the nematicidal effect still remained high (96.67%). These data provide evidence that the antinematode agents may not be enzymes (chitinase and protease); thus, the agents of interest may be small natural compounds with high thermal stability [2,4]. Considering this, we used typical methods for the extraction and purification of active ANCs, including solid–liquid with solvent extraction, separations via an opened silica gel column, and coupling with bioactivity testing. As a result, two active compounds were isolated and identified as thymine (**1**) [31] and hexahydropyrrolo [1,2-a] pyrazine-1,4-dione (**2**) [32] based on the analysis of $^1$H-NMR, $^{13}$C-NMR, HMBC, and HSQC and the HR-ESI-MS spectra in comparison to those of the reported compounds in previous studies. The extraction and isolation of targeting compounds are summarized in Scheme 1. The NMR and mass data of the two compounds and the comparison NMR data of these compounds relative to those of the reported compounds are summarized and presented in Table S1, Table S2, and Figures S1–S8 in the Supplementary Materials.

**Table 1.** Bioactivities of supernatants produced by *B. velezensis* RB.EK7 before and after high-temperature treatments.

|  | Chitinase Activity (IU/mL) | Protease Activity (IU/mL) | Mortality of *J2* Nematodes (%) |
|---|---|---|---|
| Supernatants no treated at high temperature | 4.5 | - | 93.67 |
| Supernatants treated at 100 °C in 60 min | - | - | 96.67 |

Before the elucidation of chemical structures and bioactivity assessment of these compounds, the purity of these targeting compounds was confirmed using UHPLC. As shown in Figure 2, these two compounds, thymine (**1**) and hexahydropyrrolo [1,2-a] pyrazine-1,4-dione (**2**), appeared as single peaks at the retention time (RT) of 3.915 min and 4.913 min, respectively, and these compounds were also confirmed to be of high purity grade of approximately 100%; as such, these molecules were qualified for further investigation with respect to their nematicidal activity.

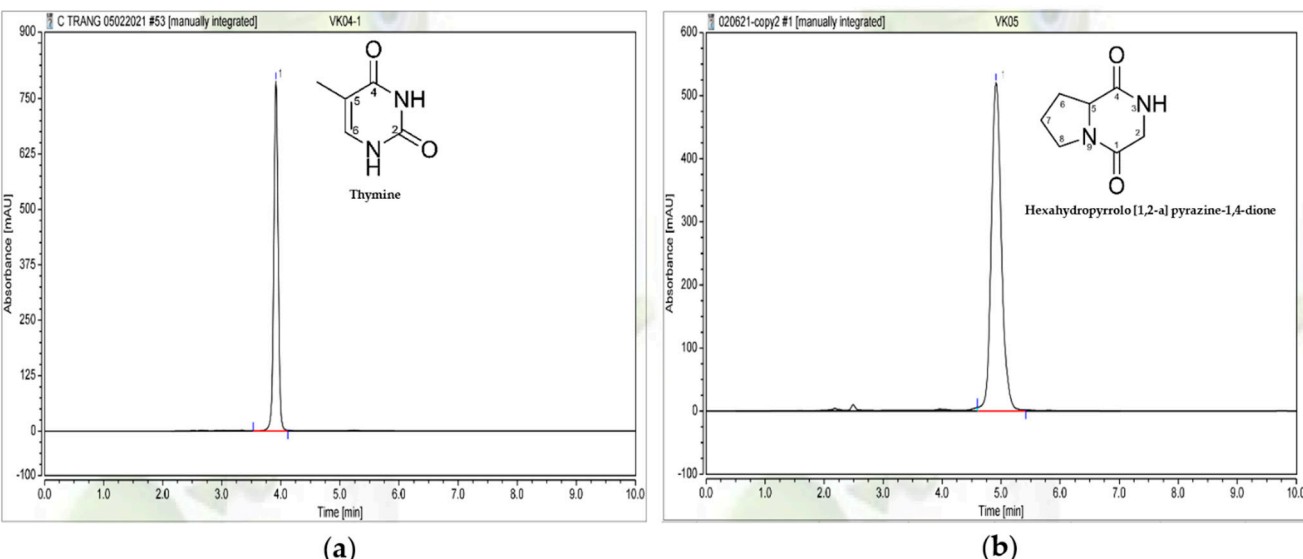

**Figure 2.** The HPLC profiles of the two antinematode compounds, including thymine (**a**) and hexahydropyrrolo [1,2-a] pyrazine−1,4−dione (**b**) purified and identified in this study.

Hexahydropyrrolo [1,2-a] pyrazine-1,4-dione is naturally obtained from several microbial strains such as *S. antioxidans* sp. nov. [33], *S. nigra* sp. nov. [34], *S. xanthophaeus* [35], and *B. tequilensis* [36]. However, there is no report on the production of this metabolite from the bacterium *B. velezensis.* Currently, thymine is mainly synthesized chemically by dissolving methyl methacrylate in a solvent of methanol [37]. There is also no literature available on the biosynthesis of thymine from the bacterial genus *B. velezensis.* Thus, these new evidences enrich the production of secondary metabolites from *B. velezensis* and may support the new, valuable, and natural sources of these active compounds for potential applications. In addition, this study is the first to report the reuse of organic wastes for the cost-effective production of these two compounds via microbial fermentation.

### 3.3. The Antinematode Activity of the Purified Compounds Produced by B. velezensis RB.EK7

The two purified compounds, thymine (**1**) and hexahydropyrrolo [1,2-a] pyrazine-1,4-dione (**2**), were evaluated for their nematicidal effects. Potential nematicidal candidates are considered to inhibit both *J2* nematodes and egg hatching. Thus, we evaluated the effect of these compounds against both *J2* nematodes and egg hatching. Hemi-pyocyanin, an ANC obtained in our previous work [14], was also used for the comparison. In assays for the antinematode effect, DMSO was used as a negative control and to dissolve the samples. In the negative group, the survival rate of *J2* nematodes was approximately 96.74%, and the egg-hatching rate was up to 88%; therefore, DMSO was appropriate for dissolving the samples and a suitable negative control in the experiments [14]. As shown in Figure 3a, all tested compounds showed nematicidal effects with more than 40% mortality of *J2* nematodes; in particular, thymine demonstrated slightly higher activities than the positive compound with a *J2* nematode mortality of 68% and 60% at the tested concentration of 2.5 mg/mL. At a higher treated concentration (20 mg/mL), hexahydropyrrolo [1,2-a] pyrazine-1,4-dione possessed antinematode activity, with 64.2% mortality of *J2* nematodes, while both thymine and hemi-pyocyanine displayed potent antinematode effects with a substantial *J2* nematodes mortality of 99–100%. This result indicated that thymine and hexahydropyrrolo [1,2-a] pyrazine-1,4-dione may be potential and moderate anti-*J2* candidates, respectively.

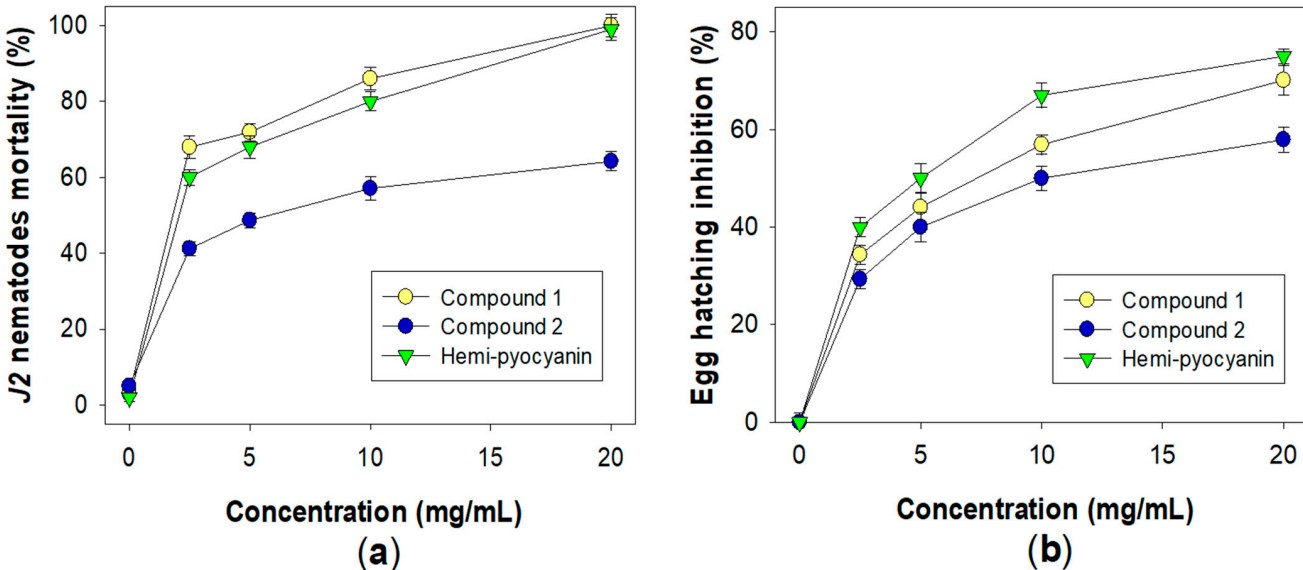

**Figure 3.** The nematicidal effects of purified thymine (compound **1**), hexahydropyrrolo [1,2-a] pyrazine-1,4-dione (compound **2**), and hemi-pyocyanin (an antinematode compound used as the positive control). The *J2* nematodes mortality (**a**) and egg-hatching inhibition effect (**b**).

The effect of these purified compounds and hemi-pyocyanin on nematode egg hatching was also detected, and the data are summarized in Figure 3b. At a low concentration (2.5 mg/mL), these compounds demonstrated inhibition against egg hatching, with inhibitory values of less than 40%. The egg-hatching inhibition effect significantly increased when higher concentrations of compounds (thymine, hexahydropyrrolo [1,2-a] pyrazine-1,4-dione, and hemi-pyocyanin), at 5, 10, and 20 mg/mL, were used, and the inhibitory values were 34.31–70.1%, 40–57.9%, and 50–75%, respectively. Similarly to *J2* nematodes mortality, thymine and hemi-pyocyanin displayed a potent effect on egg hatching, while hexahydropyrrolo [1,2-a] pyrazine-1,4-dione showed moderated activity against egg hatching. Overall, thymine may be suggested as a potential antinematidic agent, while hexahydropyrrolo [1,2-a] pyrazine-1,4-dione was found to be a moderate ANC.

Hexahydropyrrolo [1,2-a] pyrazine-1,4-dione has been shown to have several potential biological activities, including antibiotic activities against various strains of S. aureus [36] antioxidant effects [38] and algicidal against *Microcystis aeruginosa* [39]. While data on the biological effects of Thymine are not available, new biological functions notably include *J2* nematodes inhibition, and the anti-nematode egg hatching of these two purified compounds was investigated for the first time in this study. Thus, this study may provide novel potential applications with respect to thymine and hexahydropyrrolo [1,2-a] pyrazine-1,4-dione produced by *B. veleznesis* RB.EK7.

*3.4. The Action Mechenism of Antinematode Compounds via Docking Studies*

The novel nematicidal effect of thymine and hexahydropyrrolo [1,2-a] pyrazine-1,4-dione against black pepper nematodes was first reported in this study. Thus, the mechanism action models of these active compounds against nematodes are still unknown. To preliminarily predict the mechanism of action of these compounds, we assessed the molecular mechanism of anti-nematode activity via virtual screening assays [40–44]. The protein AchE is an enzyme commonly used for the inhibition of nematode *M. incognita*, a major species of the genus *Meloidogyne* [41]. Thus, this protein (AchE) was used as the targeting enzyme for the docking study.

In docking simulations, the RMSD and docking score (DS, binding energy) were considered important parameters to confirm the interaction and whether the binding energy of the ligand towards the protein enzyme is significant. When a ligand (compound inhibitor) interacts and binds to a protein enzyme with the DS and RMSD values of lower

than −3.20 kcal/mol and 3.0 Å, respectively, the compound is considered a potential inhibitor [23,24,29]. As shown in Table 2, both ligands, thymine, and hexahydropyrrolo [1,2-a] pyrazine-1,4-dione interacted with the protein AchE with RMSD and DS values in the range of 1.02 Å to 1.35 Å and −6.89 kcal/mol to −7.07 kcal/mol, respectively, which is significantly less than 3.0 Å and −3.20 kcal/mol. This result indicated that thymine and hexahydropyrrolo [1,2-a] pyrazine-1,4-dione are potent inhibitors of protein AchE; as such, the molecular mechanism action of these two compounds against nematodes may be high because of the inhibition of the protein AchE [41]. For a detailed observation of the interactions between the ligands at the active sites on AchE, the 3D structures of the total enzyme and the ligands at the active sites were examined and are presented in Figures 4 and 5. The output data of MOE (Figure 4) indicate that the thymine ligand interacts with AchE at the active sites via creating four linkages (2 H-acceptors, 1H-donor, and 1 pi-H) with some prominent amino acids, including Asp182 (3.20/-0.9/1H-donor), Lys51 (2.99/-2.3/H-acceptor), Asn183 (3.18/-1.7/H-acceptor), and Trp179 (4.30/-1.1/pi-H), while the hexahydropyrrolo [1,2-a] pyrazine-1,4-dione ligand interacts with the protein AchE at the active sites through three linkages (2H-donor and 2 H-acceptor) via connecting with three amino acids at the active sites, including Met175 (3.87/-0.8/H-donor), Phe35 (3.05/-2.8/H-donor), and Lys51 (3.11/-1.3/H-acceptor) (Figure 5).

**Table 2.** The docking study results of the interaction of the two ligands with the target enzyme acetylcholinesterase (AchE).

| Ligands. (Inhibitors) | Symbol (Ligand-Protein) | RMSD (Å) | DS (kcal/mol) | Number of Interactions | Amino Acids Interacting with the Ligand [Distance (Å)/E (kcal/mol)/Linkage Type] |
|---|---|---|---|---|---|
| Thymine (TM) | TM-AchE | 1.35 | −7.07 | 4 linkages (2 H-acceptors, 1H-donor, and 1 pi-H) | Asp182 (3.20/-0.9/1H-donor) Lys51 (2.99/-2.3/H-acceptor) Asn183 (3.18/-1.7/H-acceptor) Trp179 (4.30/-1.1/pi-H) |
| Hexahydropyrrolo [1,2-a] pyrazine-1,4-dione (HP) | HP-AchE | 1.02 | −6.89 | 3 linkages (2H-donor and 2 H-acceptor) | Met175 (3.87/-0.8/H-donor) Phe35 (3.05/-2.8/H-donor) Lys51 (3.11/-1.3/H-acceptor) |

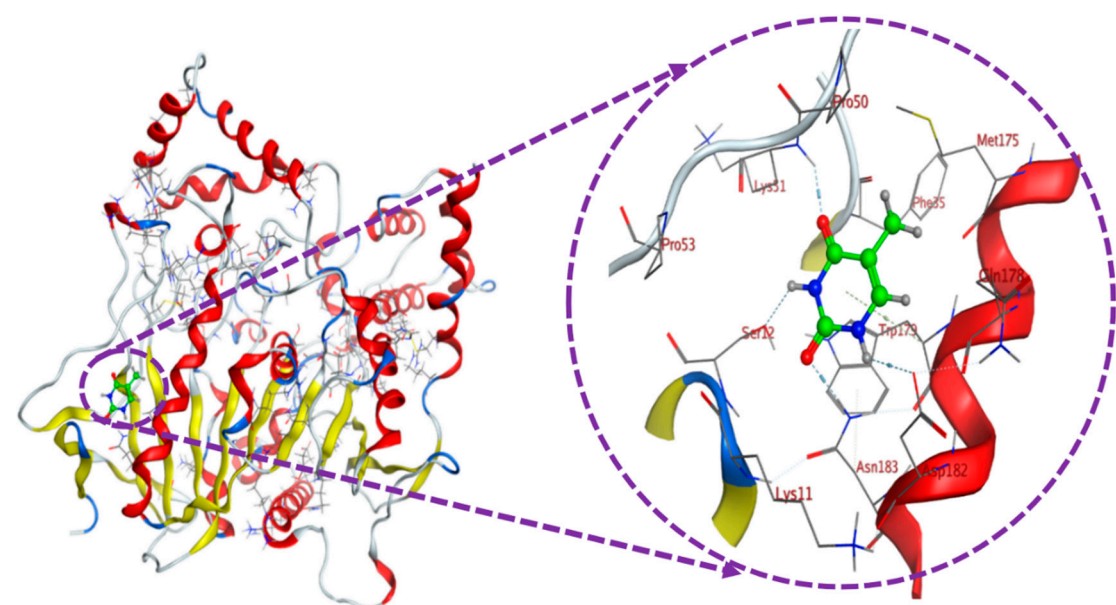

**Figure 4.** Interaction of thymine with enzyme acetylcholinesterase-targeting antinematode effects at the active site.

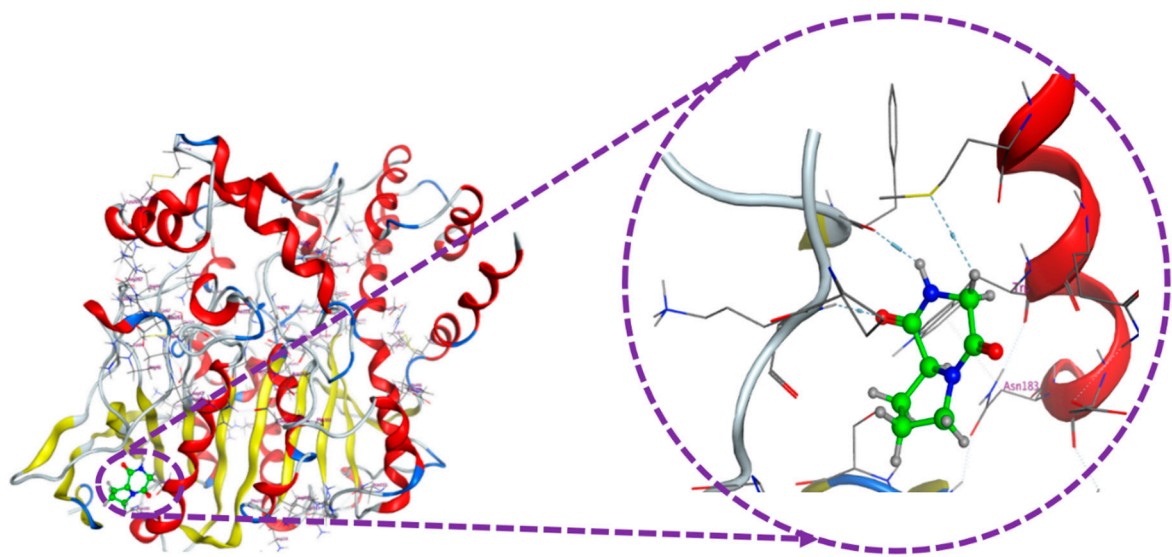

**Figure 5.** Interaction of hexahydropyrrolo [1,2-a] pyrazine-1,4-dione with enzyme acetylcholinesterase-targeting antinematode effects at the active site.

The above virtual study results indicated that the potent antinematode activity of these purified compounds may possibly be due to the inhibition of the targeting enzyme AchE. To confirm this molecular mechanism, these two compounds were tested for their effects against AchE using the assay presented in our previous study [25]. As shown in Figure 6, these compounds demonstrated AchE inhibitory activities with inhibition values of more than 70% at the tested concentration of 2.5 mg/mL, and they display the highest inhibition values (approximately 94%) at the tested concentration of 10 mg/mL. Based on the in vitro bioactivity of nematicidal effects and the virtual study of the two compounds, it suggested that thymine and hexahydropyrrolo [1,2-a] pyrazine-1,4-dione could be a good candidate for the management of black pepper nematodes. However, further studies, such as the effect of thymine and hexahydropyrrolo [1,2-a] pyrazine-1,4-dione on *J2* nematodes, egg hatching, microbiota in cultivated soil, black pepper seedlings, black pepper trees in the greenhouse, and in-field conditions should be further examined.

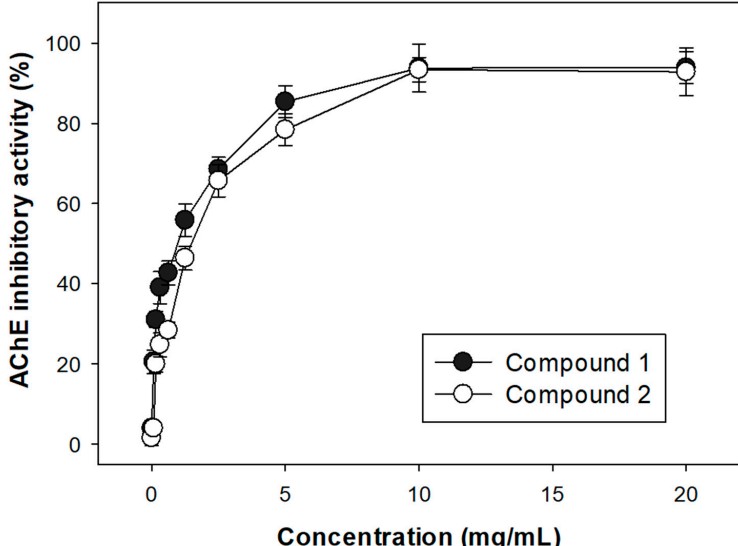

**Figure 6.** Acetylcholinesterase (AChE) inhibitory activity of purified thymine (compound **1**) and hexahydropyrrolo [1,2-a] pyrazine-1,4-dione (compound **2**).

## 4. Conclusions

Almost all organic wastes examined in this study were found as potent substrates for fermentation by *B. veleznesis* RB.EK7 to produce ANCs. Of these substrates, SSP was found to be the most effective in enhancing the production of ANCs during fermentation. Thus, the process of fermentation for the production of ANCs from organic wastes was established under suitable conditions, with the medium containing 0.8% SSP, an initial pH of 6.5–7.0, cultivation temperature of 35–37 °C, shaking speed of 150 rpm, and cultivation time of 72 h. Two active compounds were isolated from the supernatant and identified as thymine and hexahydropyrrolo [1,2-a] pyrazine-1,4-dione and demonstrated as novel potent and moderate ANCs. The virtual analysis coupling with the experimental enzyme inhibition results indicated that the potent antinematidic effect of these two compounds may be highly possible due to the inhibition of the target enzyme acetylcholinesterase.

**Supplementary Materials:** The following supporting information can be downloaded at: https://www.mdpi.com/article/10.3390/agronomy12102300/s1, Table S1: The comparison of the NMR data of compound **1** to those of the reported compound; Table S2: The comparison of the NMR data of compound **2** to those of the reported compound; Figure S1: $^1$H NMR spectrum of compound **1** (thyamine), measured in DMSO-d6 at 500 MHz; Figure S2: $^{13}$C NMR spectrum of compound **1** (thyamine), measured in DMSO-d6 at 125 MHz; Figure S3: HR-ESI-MS spectrum of compound **1** (thyamine): [M-H]$^-$ at *m/z* 125.0354; Figure S4: $^1$H NMR spectrum of compound **2** (hexahydropyrrolo [1,2-a] pyrazine-1,4-dione), measured in DMSO-d6 at 500 MHz; Figure S5: $^{13}$C NMR spectrum of compound **2** (hexahydropyrrolo [1,2-a] pyrazine-1,4-dione), measured in DMSO-d6 at 125 MHz; Figure S6: HSQC spectrum of compound **2** (hexahydropyrrolo [1,2-a] pyrazine-1,4-dione); Figure S7: HSQC spectrum of compound **2** (hexahydropyrrolo [1,2-a] pyrazine-1,4-dione); Figure S8: HR-ESI-MS spectrum of compound **2** (hexahydropyrrolo [1,2-a] pyrazine-1,4-dione).

**Author Contributions:** Conceptualization, V.B.N., T.H.T.T., S.-L.W. and A.D.N.; methodology, T.H.T.T. and V.B.N.; software, V.B.N., T.Q.P. and M.D.D.; validation, S.-L.W., T.H.T.T. and A.D.N.; formal analysis, V.B.N., T.Q.T., T.P.H.T. and T.A.H.L.; investigation, T.H.T.T. and T.H.N.; resources, V.B.N., T.H.T.T., S.-L.W. and A.D.N.; data curation, S.-L.W., V.B.N., T.H.T.T. and A.D.N.; writing—original V.B.N. and T.H.T.T.; writing—review and editing, V.B.N. and S.-L.W.; visualization, V.B.N.; supervision, A.D.N. and V.B.N.; project administration, T.H.T.T., V.B.N., and S.-L.W. All authors have read and agreed to the published version of the manuscript.

**Funding:** This research was funded by a grant from Tay Nguyen University (T2022-40CBTĐ) and supported in part by the Ministry of Science and Technology, Taiwan (MOST 111-2320-B-032-001; MOST 110-2320-B-032-001; MOST-110-2923-B-032-001).

**Data Availability Statement:** Not applicable.

**Conflicts of Interest:** The authors declare no conflict of interest.

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
