# Peer review of "Novel Nematocidal Compounds from Shrimp Shell Wastes Valorized by Bacillus velezensis RB.EK7 against Black Pepper Nematodes"

_agronomy, doi:10.3390/agronomy12102300_

Round 1

Author Response

Detailed response to reviewers' comments

 Manuscript ID: agronomy-1865524The Original Title: “Novel Black Pepper Antinematode Compounds Produced by Bacillus velezensis RB.EK7 Conversion of Organic Wastes

Dear Reviewer/Advancer

We feel pleasure to thank you for your time and effort, as well as your excellent suggestions for refining the readability and impact of our manuscript. We have gone through all the suggestions cautiously and made the revisions accordingly; and all revised parts have been typed in red in the revised manuscript. Finally, we would like to express our deep thanks to your comments and suggestions again. You certainly have served to improve the quality of this paper. We hope our response may meet the satisfactory.

Looking forward to hearing from you.         

Thanking you,

Yours Sincerely,

San-Lang Wang

Professor, Life Sciences Development Center/Department of Chemistry, Tamkang University, Taiwan

Detailed response Reviewer #1:

Comments and Suggestions for Authors:

In this manuscript, the author produce and purify the chemical structures of

antinematode compounds from Bacillus veleznesis RB.EK7 which would promote the

management of black pepper nematodes in agriculture. However before it is

considered for acceptance, the manuscript needs major revision and the following

issues should be addressed.

(1) Anti - J2 nematodes should be percentage of corrected mortality.

Reply: this item was revised accordingly the comments in the text and Figure 5a

(2) The corrected egg hatching rate should be calculated.

Reply: The effect of the compounds (thymine, hexahydropyrrolo [1,2-a] pyrazine-1,4-dione, and hemi-pyocyanin) on nematode egg hatching was expressed “egg hatching inhibition” value instead of “The egg hatching rate”. This items was revised in the text and in the figure 5b.

(3) In line 54, recent study indicated that 2-furoic acid of fungal origin could serve as

a preferable lead structure for novel nematicide research and development. Authors

may wish to review and supplement the following research paper:

(Wang L, Qin Y, Fan Z, Gao K, Zhan J, Xing R, Liu S, Li P. Novel Lead Compound

Discovery from Aspergillus fumigatus 1T-2 against Meloidogyne incognita Based on

a Chemical Ecology Study. J Agric Food Chem. 2022 Apr 20;70(15):4644-4657. )

Reply: Thank for your recommend valuable reference concerning microbial antinematode compounds. This information was added in the introduction section for enriching the information.

(4) The docking study results indicated that the potent antinematode effect of these

compounds may be possibly due to the inhibition of the targeting enzyme AchE. The

authors need to supplement the evidences to prove that these compounds could

inhibite the activity of AchE by enzyme activity test.

Reply: The AchE inhibitory activity of these compounds were tested and added in the manuscript (in the section: “3.3. The Antinematodes Activity of the Purified Compounds Produced by Bacillus velezensis RB.EK7”). The Achetylcholinesterase Inhibition Assay was also added in the section 2.2.6.

(5) There are some grammar mistakes in the manuscript. The manuscript should be

carefully revised.

Reply: we already checked and revised according to the comments. One more time, we would like to pay our deeply thanks for your very careful review, and valuable comments for enhancement of the quality of the manuscript.

Reviewer 2 Report

Novel Black Pepper Antinematode Compounds Produced by Bacillus velezensis RB.EK7 Conversion of Organic Wastes

The topic is interesting; it deals with valorizing waste to produce new compounds with good effectiveness, but some mistakes that must be considered.

Title:

I suggest rephrasing it to “Novel nematocidal compounds from shrimp shell wastes valorized by Bacillus velezensis RB.EK7 against black pepper nematodes

Abstract need to be reformulated regarding adding relevant results, an abbreviation such as (J2), and logistical and structural errors 

Keywords, each keyword should be short

Line 40, delete “of these countries”

Line 60 indicates the abbreviation

Producing nematocidal compounds from waste fermentation needs to be clearer in the introduction and discussion sections with recent citations. Try using: 

https://doi.org/10.1016/j.sjbs.2022.01.004, https://doi.org/10.1016/j.sjbs.2021.10.013 , DOI: 10.1016/j.sjbs.2021.08.035

Line 86, scientific names must be italic

Line 95 10000xg

Unify the speed of rotation unit xg or rpm

Reformulate the design of scheme 1 and replace figure 1 with scheme 1, then adjust the number of all figures

Sec 2.2.2, 

 The head of the section reformulated to “nematocidal activity.”

Lines 161 and 166 “Two hundred…..” enter as a new paragraph 

Line 164, the origin of this device

The experiments lack the control

Where the experimental design 

Figure 2 into scheme 2\

Sec 2.2.3. reformulate all subtitles

Reduce citation in line 228

Line 240, 243 adjust temperature unit °C and throught the manuscript

Line 267 should be figure 1

Table 1 includes the text

Reformulate the data of NMR structure for clarity and ease for readers

Reformulate the title of 3.4

Check language by expert

Check the outputs of all references and delete the old ones.

Author Response

Manuscript ID: agronomy-1865524The Original Title: “Novel Black Pepper Antinematode Compounds Produced by Bacillus velezensis RB.EK7 Conversion of Organic Wastes

Dear Reviewer/Advancer

We feel pleasure to thank you for your time and effort, as well as your excellent suggestions for refining the readability and impact of our manuscript. We have gone through all the suggestions cautiously and made the revisions accordingly; and all revised parts have been typed in red in the revised manuscript. Finally, we would like to express our deep thanks to your comments and suggestions again. You certainly have served to improve the quality of this paper. We hope our response may meet the satisfactory.

Looking forward to hearing from you.         

Thanking you,

Yours Sincerely,

San-Lang Wang

Professor, Life Sciences Development Center/Department of Chemistry, Tamkang University, Taiwan

Detailed response to reviewers' comments

Reviewer #2:

Novel Black Pepper Antinematode Compounds Produced by Bacillus velezensis RB.EK7 Conversion of Organic Wastes

The topic is interesting; it deals with valorizing waste to produce new compounds with good effectiveness, but some mistakes that must be considered.

Title:

I suggest rephrasing it to “Novel nematocidal compounds from shrimp shell wastes valorized by Bacillus velezensis RB.EK7 against black pepper nematodes

Reply: Thanks so much for your very positive comments and have interesting view on our work. The tittle was modified according to the comment.

Abstract need to be reformulated regarding adding relevant results, an abbreviation such as (J2), and logistical and structural errors

Reply: Abstract section was revised according to the suggestion.

Keywords, each keyword should be short

Reply: The Keywords were modified.

Line 40, delete “of these countries”

Reply: “of these countries” was deleted.

Line 60 indicates the abbreviation

Reply: this item was revised.  

Producing nematocidal compounds from waste fermentation needs to be clearer in the introduction and discussion sections with recent citations. Try using:

https://doi.org/10.1016/j.sjbs.2022.01.004, https://doi.org/10.1016/j.sjbs.2021.10.013, DOI: 10.1016/j.sjbs.2021.08.035

Reply: Thank for your recommend valuable references concerning management of nematodes. This information was added in the introduction section for enriching the information and the producing nematocidal compounds from waste fermentation was also discussed.  

Line 86, scientific names must be italic

Reply: We already checked through the manuscript and revised these items.

Line 95 10000xg

Unify the speed of rotation unit xg or rpm

Reply: Thanks for reminding us, we used rpm to unify the speed of rotation unit.

Reformulate the design of scheme 1 and replace figure 1 with scheme 1, then adjust the number of all figures

Reply: This items were revised accordingly the comments.   

Sec 2.2.2,

The head of the section reformulated to “nematocidal activity.”

Reply: The head of the section 2.2.2. was reformulated as “Nematocidal activity assays”.

Lines 161 and 166 “Two hundred…..” enter as a new paragraph

Reply: This items were revised accordingly the comments.   

Line 164, the origin of this device

The experiments lack the control

Where the experimental design

Reply: The above items were revised. This section (2.2.3) is not an experimental design, it only was used for testing nematicidal activity (like an indicator). The experimental designs, including section 2.2.1. (these experiments were designed and conducted for selecting the best C/N souces, and establish the suitable fermentation process for enhancing antinematode activity of the supernatant), section 2.2.2. (these experiments were conducted for isolation and identification of chemical structures of targeting nematicidal compounds), and section 2.2.5. was designed for preliminarily prediction of the mechanism of action of these compounds, we assessed the molecular mechanism of anti-nematode activity via virtual screening assays).    

Figure 2 into scheme 2\

Reply: Scheme 2 was used instead of Figure 2.

Sec 2.2.3. reformulate all subtitles

Reply: all subtitles of this section was reformulated.

Reduce citation in line 228

Reply: The citation was reduced..

Line 240, 243 adjust temperature unit °C and throught the manuscript

Reply: This item was revised.  

Line 267 should be figure 1

Reply: it was done

Table 1 includes the text

Reply: Table 1 was added in the text.

Reformulate the data of NMR structure for clarity and ease for readers

Reply: For more clear, we add the tables (Table S1 and Table S2) of the comparison NMR data of these compounds to those of the reported compounds, and all the data related NMR and mass spectra data were moved to the supplementary section for briefly and easier for readers. Thanks for your comments.   

Reformulate the title of 3.4

Check language by expert

Check the outputs of all references and delete the old ones.

Reply: The above items were done. One more time, we would like to pay our deeply thanks for your very careful review, and valuable comments for enhancement of the quality of the manuscript

Round 2

Reviewer 1 Report

This manuscript has been revised according to the suggestions of reviewers, and all the questions have been answered. This manuscript is acceptable for publication.

Reviewer 2 Report

Accepted in agronomy